# Physical Activity and Diet in Older Women: A Narrative Review

**DOI:** 10.3390/jcm12010081

**Published:** 2022-12-22

**Authors:** Anna Vittoria Mattioli, Valentina Selleri, Giada Zanini, Milena Nasi, Marcello Pinti, Claudio Stefanelli, Francesco Fedele, Sabina Gallina

**Affiliations:** 1Istituto Nazionale per le Ricerche Cardiovascolari, 40126 Bologna, Italy; 2Department of Medical and Surgical Sciences for Children and Adults, University of Modena and Reggio Emilia, 41124 Modena, Italy; 3Department of Life Sciences, University of Modena and Reggio Emilia, 41125 Modena, Italy; 4Surgical, Medical and Dental Department of Morphological Sciences Related to Transplant, Oncology and Regenerative Medicine, University of Modena and Reggio Emilia, 41124 Modena, Italy; 5Department of Quality of Life, Alma Mater Studiorum, 40126 Bologna, Italy; 6Department of Clinical, Internal, Anesthesiology and Cardiovascular Sciences, Sapienza University of Rome, 00161 Rome, Italy; 7Department of Neuroscience, Imaging and Clinical Sciences, “G. D’Annunzio” University, 66100 Chieti, Italy

**Keywords:** aging, women, nutrients, physical activity, long COVID, monitoring, vital signs

## Abstract

Physical activity and diet are essential for maintaining good health and preventing the development of non-communicable diseases, especially in the older adults. One aspect that is often over-looked is the different response between men and women to exercise and nutrients. The body’s response to exercise and to different nutrients as well as the choice of foods is different in the two sexes and is strongly influenced by the different hormonal ages in women. The present narrative review analyzes the effects of gender on nutrition and physical activity in older women. Understanding which components of diet and physical activity affect the health status of older women would help target non-pharmacological but lifestyle-related therapeutic interventions. It is interesting to note that this analysis shows a lack of studies dedicated to older women and a lack of studies dedicated to the interactions between diet and physical activity in women. Gender medicine is a current need that still finds little evidence.

## 1. Introduction

Physical activity and diet are essential for maintaining good health and preventing the development of non-communicable diseases [1,2,3]. As a response to population ageing, different conceptual models such as active ageing, successful ageing or healthy ageing have been proposed to address the notion of ageing well [4]. However, women are less likely to engage in sports and physical activity due to high social pressures and stereotypes that force women to play numerous time-consuming roles in family, society and work [5,6]. This attitude in women has worsened during the recent pandemic leading to increase sedentary behaviours and subsequent loss of muscle [5,6,7]. During the global pandemic older adults’ access to programs and services to facilitate the adoption of healthy lifestyles, such as gyms has been disrupted [8]. As a result, web-based interventions may represent a solution for helping older adults adopt and maintain a healthy lifestyle [8] The beneficial role of regular physical activity and structured exercise in health of older adults has been suggested by the World Health Organization [9]. The American College of Sports Medicine supported this suggestion in its guidelines published in 2022 [10]. More recently some manuscript underlined the popularity of fitness programs for older adults in the health and fitness industry at European and global level [11,12]. Women tend to have a longer life expectancy, however, while they live longer, women generally have worse health and more chronic diseases than men [13,14]. As a response to population ageing, different conceptual models such as active ageing, successful ageing or healthy ageing have been proposed to address the notion of ageing well [15].

Since physical activity and nutrition are directly related to the risk of chronic disease, it is essential to understand how these modifiable factors are related to health. The relationship between nutrition and health varies throughout life, with women having unique nutritional needs based on physiological and hormonal changes at various life times (e.g., menstruation, pregnancy, breastfeeding, menopause) [16]. However, women can experience deficiencies in their nutrition and micronutrient intake during periods of hormonal change in their lives. These deficiencies can affect aspects of your lifestyle such as sleep patterns and general quality of life and, in the long term, can impact your risk of developing chronic diseases [16,17,18].

Furthermore, physical activity must be associated with a balanced energy intake and the choice of the right foods to achieve the expected results and preserve health. It is conceivable that the choice of foods to be taken in subjects who perform regular physical activity is different in women and in men [13,19]. It is well known that the absorption and bioavailability of nutrients are different between women and men and this must influence the choice of diet at different stages of life [16,20]. The favorable effects of diet on human health are mediated by various nutrients, among these phenolic compounds are among the most widespread and well-known phytochemicals in plants and are found in many foods and drinks, such as fruit, vegetables and chocolate, as well as in coffee, tea, beer and wine [20,21]. Phenolic compounds of plant origin influence health through an antioxidant and anti-inflammatory action, this effect is different for men and women [20,21]. It is interesting to note that most of the studies analyze the differences in the choice of foods in relation to the different stages of life: young, adult and older adults. Whereas, only a few manuscripts have analyzed gender differences in dietary choice, although it is known that the different hormonal phases in women mainly influence the exposure to the risk of chronic diseases and that diet is a determining factor in the development of chronic non-communicable diseases [5,22,23].

## 2. Methods

Our search strategy was designed to inform this Review relating to gender differences in diet and physical activity in older healthy subjects. We searched MEDLINE, Scopus and Web of Science. In brief, we used a combination of terms relating to older adults (e.g., “elderly women”, “older women”, “older adult” and “longevity”) and lifestyle habits (e.g., “food”, “nutrients”, “physical activity”). In all these combinations the terms “women”, “sex”, “gender” and “diet” were mandatory. For studies to be included in this Review, they had to: (i) report on primary research, (ii) be published in peer-reviewed journals, (iii) be written in English and include data on gender differences analysis, or on factors associated with wellbeing in older women. Due to the fact that very little data are available we included papers published in the last 10 years.

The present narrative review briefly analyzes the effects of gender on nutrition and physical activity in older women and suggests future prospective for prevention of chronic disease in women. Understanding which components of diet and physical activity affect the health status of older women would help target non-pharmacological but lifestyle-related therapeutic interventions.

## 3. Results

Among 632 citations obtained at November 2022, articles that were considered to contain the most important and novel data on the effects of gender on nutrition and physical activity in older women were included in this narrative review. Due to the very limited number of manuscripts that focused on older women we included also manuscript that analyzed gender differences in adult subjects. The analysis of original manuscript (excluding review) showed that the great majority of them focused on vitamin D (101 manuscript) and sarcopenic obesity or obesity (57 manuscript). Only few manuscripts analyzed the relationship between food and physical activity in older adult (56 original manuscript) and only 16 included a specific analysis by sex or gender. The great majority of studies analyzed middle-aged women or pre-menopausal women and included a wide range of age.

## 4. Discussion

### 4.1. Vitamin D Deficiency

The first aspect that emerges in the studies carried out on older women is the objective of evaluating the concentration of vitamin D with the women’s health status. A low vitamin D status is a very common condition worldwide, and several studies from basic science to clinical applications have highlighted a strong association with chronic diseases, as well as acute conditions [24]. Estimates of the prevalence of 25 (OH)D levels < 50 nmol/L (or 20 ng/mL) have been reported as 24% (US), 37% (Canada), and 40% (Europe) [25,26]. However, the concentrations of Vit D undergo important variations in relation to sun exposure, skin color, and absorption capacities which are influenced by the intestinal microbiota. Furthermore, it may vary by age, with lower levels in childhood and the older adults [25,26], and also ethnicity in different regions, for example, European Caucasians show lower rates of vitamin D deficiency compared with nonwhite individuals [26].

A very recent manuscript evaluated whether the role of physical activity and vitamin D in sarcopenia, obesity, and sarcopenic obesity was different between men and women. They found that low physical activity was significantly associated with higher odds of sarcopenia in women only (OR  =  1.70, 95% CI:1.18, 2.46, *p*  <  0.01). Vitamin D deficiency was only associated with sarcopenia in men (OR  =  1.85, 95% CI: 1.27, 2.69, *p*  <  0.01). Low physical activity was significantly associated with obesity, sarcopenia, and sarcopenic obesity only in participants with serum 25 (OH)D  <  20 ng/mL. This manuscript by Jia S et al. underline some differences between men and women about the role of vitamin D and PA in obesity and sarcopenia. Authors also underlined that the relationship between PA and sarcopenia was modified by serum vitamin D status. These findings highlighted the need to supplement vitamin D in individuals with physical inactivity and provide different interventions strategies to sarcopenia in men and women [27].

Many peri- and postmenopausal women are deficient in or have low levels of vitamin D and magnesium. Vitamin D is essential for bone health, as well as preventing muscle weakness, protecting against falls, and providing immune support [28]. Usually, women are more likely to take dietary supplements than men, but when we analyze the use of supplements and sports drinks in people physically active this difference disappears [16]. However, no studies have analyzed the differences between physically active women and men in response to supplement components. This gap could be responsible for the lower efficacy of supplements and reflect on less health care in women [9]. Recently, Cui et al. address the various issues related to the complex relationship between food and sports with attention to the nutrients necessary to perform sports in the best physical conditions and to obtain optimal results [19]. The relevant point explored by Cui et al. was the roles of each nutritional component that can be divided into: the protection of articular cartilage, improving muscle quality, regulating endocrine, weight control, prevention of anemia, increasing energy storage and utilization, and enhancing immune function [19]. These functions are extremely important in the older adults and must be considered when addressing the diet in older adults by specifying the differences that exist between women and men.

### 4.2. Sarcopenia and Sarcopenic Obesity

The second topic that dominates the analysis of studies dedicated to adult women is sarcopenia and sarcopenic obesity (Figure 1). Although the role of physical exercise in the prevention of fractures is recognized, there are still few studies that correlate nutrition and physical activity with respect to sarcopenic obesity and bone fragility. The UK Women’s Cohort Study investigated associations between food and nutrient intakes and hip fracture risk in, evaluated the role of body mass index (BMI) as a potential effect modifier. The study included 26,318 UK women, ages 35–69 years, and 822 hip fracture cases were identified. Results of the study suggest that the potential roles of some foods and nutrients in hip fracture prevention. Particularly protein, tea and coffee seem to play a role in prevention of hip fractures in underweight women. Specifically, every additional cup of tea or coffee per day was associated with a 4% lower risk of hip fracture (HR (95% CI): 0.96 (0.92, 1.00)). A 25 g/day increment of dietary protein intake was also associated with a 14% lower risk of hip fracture (0.86 (0.73, 1.00)) [29].

This study has the merit of having highlighted a specific role of nutrients in the prevention of fractures in older women beyond the concentration of Vit D. A decisive role on the development of fractures and also on cardiovascular risk is attributed to sarcopenic obesity. Sarcopenia is a geriatric syndrome, characterized by progressive decline in muscle strength and generalized loss of skeletal muscle mass [30]. The sarcopenic muscles display heterogeneity in fiber size, atrophy of type 2 (fast-twitch) myofibers, accumulation of intramuscular fat and connective tissue with a decreased oxidative capacity. All these conditions lead to an age-related decline in functions. Moreover, during sarcopenia there is also the loss of satellite cell, which compromises the recovery capacity of sarcopenic muscles in response to injury [31]. During adulthood, satellite cells main function is to sustain skeletal muscle regenerative capacity. The consequence of satellite cell aging is the loss of skeletal muscle regenerative capacity, which is pronounced in the sarcopenic muscle. The aging environment leads to the accumulation of cellular stressors that culminate in irreversible changes to the satellite cell [31]. For this reason, “inflammaging”, a local and a systemic chronic low-grade inflammation that arises with aging, may contribute to muscle decline by impairing stem cell function and accelerating cellular senescence. Also obesity is recognized as a state of chronic inflammation with increased circulating pro-inflammatory cytokines tumour necrosis factor (TNF)-α, interleukin (IL)-1β and IL-6 [32,33]. In animal models of chronic, local or systemic inflammation, with high levels of IL-6 and TNF-α respectively, satellite cell proliferation decreases [34] and skeletal muscle becomes atrophic [35]. It is possible that in chronic inflammation the normal coordination between macrophages and muscle satellite cells is impaired and contributes to impaired satellite cell function.

Pro-inflammatory cytokines have been found to affect gene expression of satellite cells and muscle regeneration, contributing to age- and obesity-dependent decline in muscle function. During women midlife there are drastic hormonal changes due to ovarian aging and the consequent onset of menopausal. This transition phase includes a decline in the estradiol serum concentration and elevation of follicle-stimulating hormone (FSH) levels. Hormonal changes start approximately 5 years before and continue years after the final menstrual period. Muscle and bone mass decline with aging, increasing the risk for sarcopenia and osteoporosis in later life. Both these conditions are tightly related to aging and estrogen depletion and consequently to menopausal transition [36]. A few studies also suggest that menopausal hormonal changes influence the decline in lean mass (LM) among middle-aged women. Hormonal changes seem to be the major contributors to the changes in muscle and bone tissue in women undergoing menopausal transition. The preventive role of estrogens in cardiovascular disease (CVD) may depend not only on their role in the regulation of body fat distribution but also on their antioxidant effect [37,38]. Also changes in body composition, including an increase in total adiposity and a redistribution of fat with an increase in abdominal/visceral fat accumulation occur during the transition to menopause. Fat redistribution is reflected in the presence of intermuscular adipose tissue (IMAT) [39,40]. IMCLs are found in mitochondria where they increase ROS formation, resulting in the apoptosis/autophagy of muscle cells. Thus, it is considered as one of the potential mechanisms of obesity-mediated sarcopenia pathogenesis [41]. IMAT releases pro-inflammatory cytokines resulting in muscle local inflammation. Moreover, postmenopausal women have large amount of non-contractile muscle tissue, such as intramuscular fat, compared to younger women [42]. For these reasons, IMAT is a significant predictor of both muscle and mobility function in older adults and the relationship of increased levels of IMAT and decreased strength and muscle quality is reported in several studies [43,44,45].

Adipose tissue is the major metabolic and endocrine organ, containing adipocytes but also nerve tissue, connective tissue, and immune cells, such as resident eosinophils, Breg cells, CD4 T cells, Treg cells, iNKT cells, and M2 macrophages which balance local inflammation [46,47,48,49]. Recent studies have shown that most endocrine factors associated with muscle aging, such as sex steroids, glucocorticoids and thyroid hormones may regulate some muscle mitochondrial processes, including mitochondrial quality control (MQC) pathways, OXPHOS activity, redox balance, and apoptotic signaling [50,51,52]. The negative effect of chronically elevated levels of inflammatory cytokines on muscle mitochondrial function may underlie the well-known association between low-grade inflammation and sarcopenia. Further investigations are needed to understand the complex interaction and relationship among mitochondrial dysfunction, inflammation and sarcopenia.

### 4.3. Physical Activity and Nutrients in Women

It has been reported that the timing of nutrient consumption can influence the metabolism in women. To date, 95% of our nutrient timing recommendations originate from studies conducted in men. The timing of nutrient consumption during exercise directly affects performance, fatigue recovery, fat oxidation and energy expenditure [53]. Interestingly, women often exercise on an empty stomach, driven by a desire to “burn fat”. However, evidence indicates that for women in particular, fasted exercise can attenuate fat oxidation [54]. Alternatively, exercising on a full stomach will result in a higher total daily energy expenditure and increased fat oxidation and, indirectly, improve body composition. A recent analysis suggests that consuming a bolus of protein before exercise, instead of consuming a bolus of carbohydrates, significantly increases energy expenditure and improves fat oxidation after exercise for aerobic exercise, the high intensity interval training and resistance training [55]. When this approach is combined with resistance training, it appears that pre-exercise nutrition may be more effective for women to see improvements in strength and lean body mass, than post-exercise nutrition [55].

Therefore, it is necessary that women adopt an adequate diet when engaging in physical activity and sports and it is mandatory to identify which nutrients are more suitable for women than for men. Furthermore, the different hormonal phases of a woman’s life further influence her state of health. It is plausible that gender differences influence the response to food in athletes and also in subjects who perform physical activity [56]. This aspect is not always addressed in studies that explore the effects of diet and food resulting in an important lack of knowledge considering that nutrition plays a fundamental role in maintaining health. In a previous manuscript we stressed the need to teach gender medicine in medical schools in order to optimize the prescription of physical activity for prevention and therapy [57].

Expanding lifespan is not associated with robust health for all during aging, and there has been a substantial increase in age-associated morbidity. The research field on healthy aging has focused to identify risk factors affecting health and quality of life and to provide evidence of effective and acceptable interventions [58]. As previously written, nutritional needs vary greatly between and within age groups and between genders, therefore, general dietary recommendations may not be optimal for the entire population. In November 2019 the WHO introduced the definition of Sustainable Healthy Diets. These are dietary patterns that promote individuals’ health and well-being; have low environmental impact; are accessible, affordable, safe and equitable; and are culturally acceptable. The aims of Sustainable Healthy Diets are to (a) promote optimal growth and development and support physical, mental, and social well-being at all life stages for present and future generations; (b) contribute to preventing all forms of malnutrition (i.e., undernutrition, micronutrient deficiency, overweight and obesity); (c) reduce the risk of diet-related NCDs; (d) support the preservation of biodiversity and planetary health [59]. Increasing age is associated with many physiological changes that increase the risk of undernutrition, affecting up to 22% of individuals, with subsequent physical and cognitive impairments, including reduced bone and muscle mass, increased frailty, diminished cognitive function and ability to care for oneself, and thus a higher risk of becoming dependent on care [60]. The mechanisms by which diet affects aging are not well understood, but it seems likely that a wide range of dietary factors counteract molecular damage (e.g., inflammation, oxidative stress and endothelial dysfunction) and mitigate associated functional changes that are induced in aging [61,62]. In addition, several studies have demonstrated the crucial role that gut microbiota plays in maintaining human health. As a model of healthy aging, centenarians have different gut microbiota from ordinary older people. The core microbiome of centenarians in various countries has shown some common characteristics, which are worth further exploration [63]. It is quite difficult to identify a specific diet that is effective in the older woman, a personalized approach is needed. A recent article by Sun and coworkers assessed the effects of whey protein (WP) or WP hydrolysate (WPH) combined with an energy-restricted diet (ERD) on weight reduction and muscle preservation in older women with overweight and obesity [64]. Weight loss is important for older adults with obesity, but a conventionally adopted low-calorie diet is likely to exacerbate the age-related sarcopenia. In older adults the mortality risks of sarcopenia may outweigh the potential benefits of weight loss [65]. Sun and co-workers found that an energy-restricted diet significantly decreased body weight and fat mass, with more noticeable results in the WPH group [64]. Tischmann L and coworkers evaluated long-term effects of soy nut consumption on vascular function and cardiometabolic risk markers in healthy older men and women [66]. They concluded that longer-term soy nut intake as part of a healthy diet improved endothelial function, and LDL-cholesterol concentrations suggesting mechanisms by which an increased soy food intake beneficially affects CVD risk in older adults [66].

Cubas-Basterrechea et al. evaluated the adherence to Mediterranean Diet (MedDiet) in older subjects and found an inverse relationship was established between adherence to the MedDiet and the prevalence of Metabolic syndrome [67]. Scientific evidence supported the beneficial effects of MedDiet consumption in older subjects related to longevity, quality of life, and disease prevention [68]. The MedDiet is rich in bioactive components such as antioxidants (vitamins C and E, among others), fibre, and phytosterols (from vegetables, fruits, legumes, nuts, whole grains, olive oil, and wine), and provide the correct balance of polyunsaturated fatty acids (omega-6 vs. omega-3) through regular consumption of fish, seafood, and nuts, along with high consumption of monounsaturated fats (e.g., olive oil) and low consumption of saturated fats (e.g., meat) [69,70]. The MedDiet has important and proven benefits in the prevention of chronic non-communicable diseases and promotes longevity. There is no evidence that any component of the diet is more effective in older women than in men. This requires further investigations from the perspective of a gender approach to disease prevention.

Many manuscripts underlined the critical role of physical activity, exercise training, and cardiorespiratory fitness (CRF) in the primary and secondary prevention of cardiovascular disease [71,72,73]. The very recent “Clinical practice statement of the ASPC” defined exercise training, as a subcategory of physical activity (PA), as any structured exercise regimen with the objective of improving or maintaining CRF, muscle strength, health, functional independence, athletic performance, or combinations thereof. Aerobic capacity or CRF is typically expressed as mLO_2_/kg/min or metabolic equivalents (METs; 1 MET = 3.5 mL/kg/min) and can be directly determined using gas-exchange measurements or estimated from the attained treadmill speed, percent grade, and duration (minutes) or the cycle ergometer workload, expressed as kilogram meters per minute [74]. Patients with higher fitness and a major CVD risk factor, such as diabetes, obesity, hypertension, or dyslipidemia, generally had a better prognosis those without these risk factors but with low fitness [75]. Kokkinos et al. analyzed a cohort of U.S. veterans and a very diverse population regardless of age, sex, race, and ethnicity, supporting the importance of CRF across various U.S. populations, with no increased risk at very high CRF [75]. However, physicians should be aware that CRF is modifiable with an increase in physical activity. It is known that individuals with low CRF levels who regularly engage in physical activity or exercise can significantly reduce the risk of mortality compared to individuals who remain physically inactive and have low CRF levels [76]. A prolonged period of sedentary behaviour and inactivity in older individuals accelerates the deterioration of skeletal muscle health, including loss of muscle mass and function. Decreased muscle mass in older adults is associated with increased mortality and reduced quality of life [77].

After the pandemic some article explore the effects of Diet and of physical activity on immune system assuming that a healthy diet and a regular physical activity contribute to a stronger immune response and a reduction in systemic inflammatory status [49,78,79]. Clinical studies underline that vitamins and folate, polysaccharides and dietary fiber, lipids, peptides, and natural polyphenols are important for the body’s immune system against viruses [78,79]. Natural polyphenols (flavonoids, phenolic acids, stilbenes, lignans) exert known anti-inflammatory, antimicrobial and antioxidant activities, have antiviral capacity, prevent digestion issues and reduce the risk of chronic diseases. Specifically, against the SARS-CoV-2 Virus, they act by inhibiting viral replication, disrupting viral spike protein and inhibiting the SARS-CoV-2 protease [80,81,82]. Reactive oxygen species play a crucial role in the inflammatory response. Therefore, compound with antioxidant properties have been used to reduce the cytokine storm induced by virus infection [79,83]. To date, antioxidant therapies are being considered to improve muscle responses in patients suffering from long COVID [84,85,86]. Interestingly, some of the long-standing COVID immunological and systemic features suggest signs of accelerated or premature aging and may aggravate pre-existing age-associated degenerative conditions, such as sarcopenia and cognitive decline [81,82]. There is currently a lack of specific treatments for long COVID. Patient management it is mainly based on symptomatic treatments and recommendations for conducting a healthy lifestyle. Several food supplements and natural bioactive substances have been tested for their potential to counter the long-COVID [85,86,87].

Global epidemiological trends demonstrate that women have almost double the lifetime rates of anxiety and depression compared to men and increased subclinical rates of symptoms of these disorders [75,85,86]. Women with obesity are more likely to become depressed and report symptoms of depression at a younger age compared to males [88,89]. Depression and stress lead to unhealthy nutritional habits such as cravings [89,90,91]. A craving for calorie-dense, energy-dense foods can lead to weight gain or obesity. Men and women experience food cravings differently. Women are much more likely to be craving for food (28% compared to 13% of men) and report negative feelings for indulging in cravings, while men are less likely to crave food and report positive feelings associated with cravings [91]. Food energy density is believed to be the strongest predictor of cravings in overweight and obese adults. An energy restriction study found that 75% of coveted foods were chocolate, salty snacks, ice cream, or sugary baked goods [92]. Consuming large quantities of energy-dense foods is not only problematic in terms of nutrient intake, but also in terms of energy balance [90,91]. Today, an increase in physical activity and a healthy diet is suggested in the recovery actions of the unhealthy lifestyle developed during COVID-19.

### 4.4. Monitoring of Vital Parameters

Another aspect that is becoming increasingly important is the autonomous and home monitoring of vital parameters. The 5 vital signs identified by the WHO are heart rate, respiratory rate, oxygen saturation, blood pressure and body temperature. Alterations of 1 or more parameters indicate disease [92]. The early detection of changes in vital signs typically correlates with faster detection of changes in the cardiopulmonary status of the patients [93]. Some vital signs can be influenced by common age-related pathologies, including hypertension, atherosclerosis, and arrhythmias. Atherosclerotic disease can further increase pulse pressure, which, in conjunction with a high resting heart rate, causes mechanical stress and damage to the endothelium. Finally, the stress response further promotes atherosclerosis. Atherosclerosis can reduce the flexibility of the arteries, contributing to the development of hypertension and the increase of blood pressure with age [92,94]. Similarly, some arrhythmias, i.e., atrial fibrillation increase with age. Monitoring heart beat allow the early diagnosis of heart rhythm abnormalities with the possibility of a more rapid and effective intervention [95,96]. Different devices can be used for the measurement of vital sign, however it is important that they are validated instruments and that provide adequate measurements. In the older adults, the control of vital parameters is important both at rest and during physical exercise.

### 4.5. Strengths and Limitations of the Study

The strength of the present literature review was to highlight a paucity of studies devoted to the analysis of diet and exercise in older women. It is known that gender medicine complains of a lack of studies dedicated to the differences between men and women and, although there has been an increase in the perception of this need, a gap still persists.

The limitation of the study is due to the difficulty of identifying articles analyzing outcomes in older women. While studies of menopausal women abound, older women are an underrepresented population in clinical trials.

## 5. Conclusions

It is the general opinion of experts and confirmed by numerous studies that good nutrition and regular physical activity are milestones for counteracting the effects of aging. However, the analysis of the recent literature does not highlight studies in which these two aspects are evaluated together in older women. Gender medicine needs an increase in studies dedicated to women who consider the different phases of hormonal transition. These aspects must be investigated in view of the increase in life span. We need to identify personalized pathways for older women to reduce the risk of chronic disease and improve the quality of life.

## Figures and Tables

**Figure 1 jcm-12-00081-f001:**
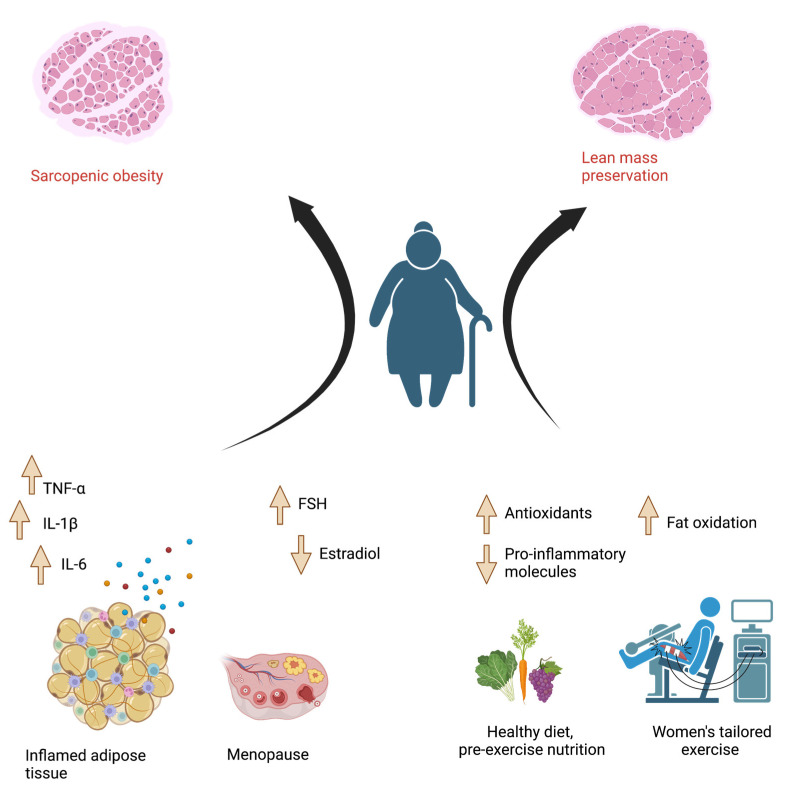
Factors that influence the development of sarcopenic obesity. Red, blue and yellow dots represent the pro-inflammatory cytokines TNF-α, IL-1β and IL-6 released by inflamed adipose tissue.

## Data Availability

Not applicable.

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
