# Peer review of "Physical Activity and Diet in Older Women: A Narrative Review"

_jcm, 2022, doi:10.3390/jcm12010081_

Round 1

Reviewer 1 Report

Dear Authors,

I have read with interest your paper. I have some observations regarding it:

- the abstract looks unstructured and short, and the use of past tense in ”performed” makes the study look like an original study, not like a review.

- in the Introduction, there is a missing reference and orthographic sign at line 30.

- there are many differences in numbering the references (before or after the ”.”, and also there are missing orthographic signs in the manuscript.

- the phrase which finishes at row 51 has no reference; also ”it is clear women can and do experience” is not a preferred way of writing.

- in my humble opinion, Figure 1 looks like an image from a high school brochure, it should be rethought.

- the term ”elderly” is nowadays changed to”older adults”. The paper starts writing only ”older adults/women”, yet in the Methods section the search did not comprise this term, but only the term ”elderly”. This could jeopardize the findings on which the review was focused.

- ”gender” and ”diet” should have been mandatorily used in the literature search.

- in the discussion, the section which properly discusses the title of the paper has around half a page, while sarcopenia is discussed in more than 2 pages

- the first part of the discussion starts abruptly with vitamin D, but this is not acknowledged with a subtitle like 3.1. ...

- it would be interesting if the authors actually find types of meals or specific diets useful for women more than men.

- furthermore, CRF should be detailed, maybe the readers are not familiarised with this physical exercise program.

- reference 64 is cited without parentheses (line 279).

- the phrase from lines 207-208 has no cited reference.

- reference 77 has the title ”Cardiovascular Effects of whole-body cryotherapy in non-professional athletes” and is written by some of the authors. This self-citation was not expected in a section of the manuscript where we expect to find the searched articles from the ”Methods” section. It looks like it is forcedly inserted in that section. 

- the other articles of Coppi F cited in the text could be relevant, even if they all discuss situations since the COVID-19 pandemic, which are not expected to be read after the Introduction part, where they are not mentioned.

- the Conclusions section states a truth that is common sense to know. The last phrase (lines 329-331) is difficult to read and understand. 

I consider all of the above when giving your manuscript ”Major Revision” status.

Good luck!

Author Response

Dear Authors,

I have read with interest your paper. I have some observations regarding it:

Question

- the abstract looks unstructured and short, and the use of past tense in ”performed” makes the study look like an original study, not like a review.

AUTHOR RESPONSE: we thank the reviewer for the observation, abstract has been rewritten

Question

- in the Introduction, there is a missing reference and orthographic sign at line 30.

AUTHOR RESPONSE: we are sorry. we have inserted a new ref #4 and update references

 Question

- there are many differences in numbering the references (before or after the ”.”, and also there are missing orthographic signs in the manuscript.

AUTHOR RESPONSE: We checked, standardized and update references

 Question

- the phrase which finishes at row 51 has no reference; also ”it is clear women can and do experience” is not a preferred way of writing.

AUTHOR RESPONSE: we agree that the sentence is confusing so we rewrote the sentence (lines 56-60 of the revised manuscript and inserted the new ref #16,17,18)  “However, women can experience deficiencies in their nutrition and micronutrient intake during periods of hormonal change in their lives. These deficiencies can affect aspects of your lifestyle such as sleep patterns and general quality of life and, in the long term, can impact your risk of developing chronic diseases.”

Question

 - in my humble opinion, Figure 1 looks like an image from a high school brochure, it should be rethought.

AUTHOR RESPONSE: as suggested by reviewer #2 we deleted figure 1.

Question 

- the term ”elderly” is nowadays changed to”older adults”. The paper starts writing only ”older adults/women”, yet in the Methods section the search did not comprise this term, but only the term ”elderly”. This could jeopardize the findings on which the review was focused.

- ”gender” and ”diet” should have been mandatorily used in the literature search.

AUTHOR RESPONSE: We thank the reviewer for the observation, actually we have used both the terms "elderly" and "older adults" in order not to loose important manuscripts in the search. As the reviewer pointed out the change in terminology is recent.

In order to clarify this point we have modified the methods as follows: “In brief, we used a combination of terms relating to elderly (eg, “elderly women”, “older women”, “older adult” and “longevity”) and lifestyle habits (eg, “food”, “nutrients”, “physical activity”). In all these combinations the term “women”, “sex” , “gender”, and “diet” were mandatory.” See lines 81-84 of the revised manuscript

Questions

- in the discussion, the section which properly discusses the title of the paper has around half a page, while sarcopenia is discussed in more than 2 pages

- the first part of the discussion starts abruptly with vitamin D, but this is not acknowledged with a subtitle like 3.1. ...

AUTHOR RESPONSE: In order to make the manuscript more balanced we have reorganized the discussion part by inserting sub-heads. The section related to food and physical activity has been expanded.

We also included a new manuscript published on november 22 that was not available when the original manuscript was send to Journal. See discussion lines 121-132 of the revised manuscript and new ref #27

[Jia S, Zhao W, Hu F, Zhao Y, Ge M, Xia X, Yue J, Dong B. Sex differences in the association of physical activity levels and vitamin D with obesity, sarcopenia, and sarcopenic obesity: a cross-sectional study. BMC Geriatr. 2022 Nov 24;22(1):898. doi: 10.1186/s12877-022-03577-4. PMID: 36434519]

Question 

- it would be interesting if the authors actually find types of meals or specific diets useful for women more than men.

AUTHOR RESPONSE: We included some studies on specific meals (see lines 279-307 of the revised manuscript) and some new refer from# 64 to 70

Question

 - furthermore, CRF should be detailed, maybe the readers are not familiarised with this physical exercise program.

AUTHOR RESPONSE: We thank the reviewer for the comment. To clarify this point we add a definition from Clinical practice statement of the ASPC. Lines 310-318 of the revised manuscript and new ref #74. We also add a comment in the Introduction section (see lines 42-47) and new ref #9-12

Questions 

- reference 64 is cited without parentheses (line 279).

- the phrase from lines 207-208 has no cited reference.

AUTHOR RESPONSE: sorry we corrected 

- Question

 reference 77 has the title ”Cardiovascular Effects of whole-body cryotherapy in non-professional athletes” and is written by some of the authors. This self-citation was not expected in a section of the manuscript where we expect to find the searched articles from the ”Methods” section. It looks like it is forcedly inserted in that section.

AUTHOR RESPONSE: Citation 77 was included as an example of vital signs monitoring, it did not emerge from the literature search but we felt it was consistent with the topic, however if the reviewer believes it is inappropriate we delete it

 Question

- the other articles of Coppi F cited in the text could be relevant, even if they all discuss situations since the COVID-19 pandemic, which are not expected to be read after the Introduction part, where they are not mentioned.

AUTHOR RESPONSE: Due to the recent pandemic, lifestyle habits especially in relation to diet, exercise and a sedentary lifestyle have been profoundly changed and we must take into account the effects this will have on the prevention of chronic diseases in the future.

Question

- the Conclusions section states a truth that is common sense to know. The last phrase (lines 329-331) is difficult to read and understand.

AUTHOR RESPONSE: We rewrote the conclusion section (see 394-402 of the revised manuscript) and we also included a section on " Strengths and limitations of the study (see lines 385-393 of the revised manuscript)

Reviewer 2 Report

General comments

 The author has clearly stated that the purpose of this narrative review was to examine the effects of physical activity and diet in older women. The paper is well-written, easy to follow and adds some merit. Given this approach, this work can enhance future attempts in similar research area. However, I have highlighted a few suggestions and concerns in my specific comments section (below) that need to be addressed before considering whether this work should be published or not.

 Specific comments

 ABSTRACT

 - The abstract is very short. Add more information regarding the summary results and conclusions. suggesting future prospective for prevention of chronic disease in women.

- The length should be approximately 200 words, aiming to attract readers to download and read the full text.

INTRODUCTION

 - I suggest adding a sentence about the beneficial role of regular physical activity and structured exercise in health of older adults according to the latest guidelines by the World Health Organization (1) and the American College of Sports Medicine (2).

- A statement about the popularity of fitness programs for older adults in the health and fitness industry at European and global level according to the latest report published by the American College of Sports Medicine (3), it could be a useful addition.

 Suggested References:

1.      Bull FC, Al-Ansari SS, Biddle S, Borodulin K, Bumanat MP, Cardonal G, et al. World Health Organization 2020 guidelines on physical activity and sedentary behaviour. Br J Sports Med 2020; 54(24): 1451-1462.

2.      American College of Sports Medicine; Liguori, G.; Feito, Y.; Fountaine, C.; Roy, B.A. ACSM’s Guidelines for Exercise Testing and Prescription, 11th ed.; Wolters Kluwer Health: Philadelphia, PA, USA, 2021.

3.      Kercher VM, Kercher K, Bennion T, Levy P, Alexander C, Amaral PC, et al. 2022 Fitness Trends from Around the Globe. ACSMs Health Fit J 2022; 26(1): 21-37.

4.      Batrakoulis, A. European survey of fitness trends for 2020. ACSM’s Health & Fitness Journal 2019; 23(6): 28–35.

METHODS

 - Inclusion and exclusion criteria for assessing eligibility of retrieved articles should be presented clearer using a numbered list such as i), ii), iii), etc.

RESULTS

 - The results section is totally missing. Results should be presented in brief in a separate section. A summary table could be also a useful and attractive approach to readers.

- Figure 1 should be improved, since it has low resolution and the creation seems truly amateur. Otherwise, it must be removed. Consider using a similar approach and quality with what you used in Figure 2. In my opinion, a graphics software program should be used (i.e., https://mindthegraph.com/app/graphical-abstract-maker?utm_source=sendinblue&utm_medium=authors&utm_campaign=email2 )

DISCUSSION

 - Strengths and limitations of the study should be presented in a separate paragraph at the end of the discussion section.

- In conclusions, you should underline the main findings and suggest future research attempts in this area while highlighting potential practical implications.

Author Response

General comments

The author has clearly stated that the purpose of this narrative review was to examine the effects of physical activity and diet in older women. The paper is well-written, easy to follow and adds some merit. Given this approach, this work can enhance future attempts in similar research area. However, I have highlighted a few suggestions and concerns in my specific comments section (below) that need to be addressed before considering whether this work should be published or not.

 Specific comments

Questions

 ABSTRACT

 - The abstract is very short. Add more information regarding the summary results and conclusions. suggesting future prospective for prevention of chronic disease in women.

- The length should be approximately 200 words, aiming to attract readers to download and read the full text.

AUTHOR RESPONSE: We thank the reviewer for the observation, abstract has been rewritten (see lines 15-25 of the revised manuscript)

Question

 INTRODUCTION 

 - I suggest adding a sentence about the beneficial role of regular physical activity and structured exercise in health of older adults according to the latest guidelines by the World Health Organization (1) and the American College of Sports Medicine (2).

 - A statement about the popularity of fitness programs for older adults in the health and fitness industry at European and global level according to the latest report published by the American College of Sports Medicine (3), it could be a useful addition.

  Suggested References: 

  1. Bull FC, Al-Ansari SS, Biddle S, Borodulin K, Bumanat MP, Cardonal G, et al. World Health Organization 2020 guidelines on physical activity and sedentary behaviour. Br J Sports Med 2020; 54(24): 1451-1462.
  1. American College of Sports Medicine; Liguori, G.; Feito, Y.; Fountaine, C.; Roy, B.A. ACSM’s Guidelines for Exercise Testing and Prescription, 11th ed.; Wolters Kluwer Health: Philadelphia, PA, USA, 2021.
  1. Kercher VM, Kercher K, Bennion T, Levy P, Alexander C, Amaral PC, et al. 2022 Fitness Trends from Around the Globe. ACSMs Health Fit J 2022; 26(1): 21-37.
  1. Batrakoulis, A. European survey of fitness trends for 2020. ACSM’s Health & Fitness Journal 2019; 23(6): 28–35.

AUTHOR RESPONSE: We thank the reviewer for her/his helpful suggestion. The introduction has been updated according to your suggestions and 4 new refs have been added (see Lines 42-47 of the revised manuscript and new ref from #9 to 12)

Questions 

METHODS 

 - Inclusion and exclusion criteria for assessing eligibility of retrieved articles should be presented clearer using a numbered list such as i), ii), iii), etc.

AUTHOR RESPONSE: We thank the reviewer for her/his helpful suggestion. Methods has been updated (see lines 80-87 of the revised manuscript). 

"In brief, we used a combination of terms relating to older adults (eg, “elderly women”, “older women”, “older adult” and “longevity”) and lifestyle habits (eg, “food”, “nutrients”, “physical activity”). In all these combinations the terms “women”, “sex”, “gender” and “diet” were mandatory. For studies to be included in this Review, they had to: i) report on primary research, ii) be published in peer-reviewed journals, iii) be written in English and include data on gender differences analysis, or on factors associated with wellbeing in older women"

Question

RESULTS

 - The results section is totally missing. Results should be presented in brief in a separate section. A summary table could be also a useful and attractive approach to readers.

AUTHOR RESPONSE: We apologize. The section results has been included (see lines 94-105 of the revised manuscript)

- Figure 1 should be improved, since it has low resolution and the creation seems truly amateur. Otherwise, it must be removed. Consider using a similar approach and quality with what you used in Figure 2. In my opinion, a graphics software program should be used (i.e., https://mindthegraph.com/app/graphical-abstract-maker?utm_source=sendinblue&utm_medium=authors&utm_campaign=email2 )

AUTHOR RESPONSE: As you suggested we deleted figure 1.

Question

DISCUSSION 

 - Strengths and limitations of the study should be presented in a separate paragraph at the end of the discussion section.

A separate paragraph on Strengths and limitations of the study has been included (see lines 385-393 of the revised manuscript)

"4.5 Strengths and limitations of the study

The strength of the present literature review was to highlight a paucity of studies devoted to the analysis of diet and exercise in older women. It is known that gender medicine complains of a lack of studies dedicated to the differences between men and women and, although there has been an increase in the perception of this need, a gap still persists.

The limitation of the study is due to the difficulty of identifying articles analyzing outcomes in older women. While studies of menopausal women abound, older women are an underrepresented population in clinical trials."

Question

- In conclusions, you should underline the main findings and suggest future research attempts in this area while highlighting potential practical implications.

AUTHOR RESPONSE:

We rewrote the conclusion section (see 394-402 of the revised manuscript)

Round 2

Reviewer 1 Report

Dear authors,

I am content with the improvement of the manuscript. 

Good luck!